# The Preventive Effect of Dietary Antioxidants on Cervical Cancer Development

**DOI:** 10.3390/medicina56110604

**Published:** 2020-11-10

**Authors:** Ayumi Ono, Masafumi Koshiyama, Miwa Nakagawa, Yumiko Watanabe, Eri Ikuta, Keiko Seki, Makiko Oowaki

**Affiliations:** 1Graduate School of Human Nursing, The University of Shiga Prefecture, Shiga 522-8533, Japan; ono.a@nurse.usp.ac.jp (A.O.); nakagawa.miw@nurse.usp.ac.jp (M.N.); ikuta.e@nurse.usp.ac.jp (E.I.); seki.k@nurse.usp.ac.jp (K.S.); owaki@nurse.usp.ac.jp (M.O.); 2Department of Women’s Health, Graduate School of Human Nursing, The University of Shiga Prefecture, Shiga 522-8533, Japan; watanabe.yu@nurse.usp.ac.jp

**Keywords:** dietary oxidant, HPV, cervical dysplasia, cervical cancer

## Abstract

Cervical cancer results from a continuous process, starting from a normal cervical epithelium after human papillomavirus (HPV) infection and progressing to cervical intraepithelial neoplasia (CIN), before finally developing into invasive squamous carcinoma (ISC). In recent decades, dietary antioxidants, such as vitamins, have received much attention in relation to cancer prevention. We reviewed the relevant literature to investigate the dietary and nutrient intake on cervical cancer. The intake of vitamins A and D and carotenoids may inhibit early cervical cancer development. The intake of folate may prevent or inhibit HPV infection rom progressing to various grades of CIN. The intake of vitamins C and E may widely inhibit the process of cervical cancer development. Polyphenols are often used in cases of cervical cancer in combination chemotherapy and radiation therapy. Regarding nutrients, different antioxidants may have differing abilities to intervene in the natural history of cervical diseases associated with HPV infection. Regarding foods, the intake of both vegetables and fruits containing multiple vitamins may widely suppress cervical cancer development. Most previous papers have described epidemiological studies. Thus, further research using in vitro and in vivo approaches will be needed to clarify the effects of the dietary and nutrient intake in detail.

## 1. Introduction

Cervical cancer remains the third most common gynecological cancer in the world and the leading cause of mortality among women in developing countries [1]. Cervical cancer develops through a continuous process, starting from the normal cervical epithelium after human papillomavirus (HPV) infection and progressing to cervical intraepithelial neoplasia (CIN), before finally becoming invasive squamous carcinoma (ISC).

Persistent high-risk human papillomavirus (HR-HPV) infection has been recognized as a necessary step in the progression to CIN, which is graded from 1 to 3 depending on the degree of epithelial abnormality, and cervical cancer. Among HR-HPVs, HPVs 16 and 18 are identified in ≥70% of cervical cancer cases [2,3]. The HPV E6 and E7 viral oncoproteins play a pivotal role in driving cells toward oncogenesis [4]. E6 can induce the formation of a complex with ubiquitin ligase and p53, resulting in the inhibition of p53-mediated apoptosis [5,6], while E7 has the ability to bind to the retinoblastoma (Rb) protein, resulting in the inhibition of Rb functions, such as cell cycle regulation [7]. 

Oxidative stress or antioxidant deficiency have frequently been reported to be associated with cervical cancer, being responsible for the production of reactive oxygen species (ROS) and ultimately causing DNA damage to cervical cells [8]. Under such conditions, cervical cells are rendered vulnerable to HPV infection and consequent cervical cancer development. 

In recent decades, dietary antioxidants, such as vitamins, have received much attention concerning their role in cancer prevention, particularly because they may prevent free-radical damage to DNA by neutralizing free radicals and oxidants enhancing the immune system and inhibiting insulin-like growth factor (IGF) [9,10,11]. 

In the present study, we reviewed the relevant literature to identify recent epidemiological studies and clinical trials in order to improve our understanding of the role of individual antioxidant nutrients at different stages of cancer development from normal cervical cells, to CINs to ISC.

## 2. Nutrients

### 2.1. Vitamin A

Vitamin A is the name of a group of fat-soluble retinoids, including retinol, retinal, and retinyl esters [12]. Beta-carotene, named provitamin A, is transformed into vitamin A by the liver, according to the body’s needs. It is the most powerful precursor of vitamin A, followed by alpha-carotene, beta-cryptoxanthin, and other carotenoids. 

Vitamin A deficiency causes oxidative stress, inhibiting the cell repair function and causing cell damage [13]. Huang et al. investigated whether or not vitamin A was independently related to HPV infection in 13,412 American women [14]. They concluded that an appropriate amount (<1448.155 μg) of dietary vitamin A may help to prevent HPV infection as it was associated with a 10% reduction in the risk of HPV infection. However, the excessive intake of dietary vitamin A (≥1448.155 μg) may increase the risk of HPV infection.

Retinol is required for the replication of basal mucosa cells and the synthesis of protein blocks [15]. Thus, retinol deficiency may be associated with an increased risk of developing squamous metaplasia and HPV infection. In a clinic-based case–control study, Yeo. et al. showed that the lowest serum retinol quartile was associated with an increased risk of developing CIN 1 in comparison to women in the highest quartile (odds ratio (OR) = 2.3) [16]. 

In a hospital-based case–control study, on the other hand, serum levels of retinol were not inversely associated with the risk of cervical cancer [17]. Therefore, retinol (vitamin A) may inhibit the early events (HPV infection and CIN 1 development) of cervical carcinogenesis [18].

### 2.2. Vitamin D

Vitamin D promotes calcium absorption in the small intestine and maintains adequate serum calcium and phosphate concentrations to enable normal mineralization of bone. Vitamin D is also needed for bone growth and bone remodeling by osteoblasts and osteoclasts [19,20]. Vitamin D has other roles, including the modulation of cell growth, the neuromuscular and immune functions, and the reduction of inflammation [20,21]. Vitamin D therefore has anti-inflammatory activity and increases the gene expression of insulin receptors and/or proteins related to the insulin-signaling cascade [22,23]. Due to its anti-inflammatory activities and effects in relation to the improvement of insulin resistance, vitamin D may be useful for ameliorating clinical and metabolic symptoms in patients with HPV infection [24].

A 2010 Japanese case–control study showed that the risk of incidental cervical neoplasia was reduced in 405 patients with an increased vitamin D intake [25]. Recently, Vahedpoor et al. performed a randomized, double-blind, placebo-controlled trial among 58 women diagnosed with CIN 1 to evaluate the effects of long-term vitamin D administration on the regression and metabolic status of the patients [26]. After 6 months of vitamin D administration, more women in the vitamin D group had regressed CIN 1 in comparison to the non-vitamin D group (84.6% vs. 53.8%, *p* = 0.01), and the intake of vitamin D led to a significant reduction in serum insulin levels in comparison to the non-vitamin D group (−5.3 ± 7.3 vs. +2.4 ± 5.9 μIU/mL, *p* < 0.001). Schlte-Uebbing et al. also reported that treatment with vitamin D vaginal suppositories (12,500 IU, three nights a week, for 6 weeks) resulted in antidysplastic effects in the CIN 1 group, but that it did not affect the CIN 2 group [27]. Ozgu et al. showed that the 25-hydroxy vitamin D level in 22 HPV-positive patients was significantly lower than that in 62 HPV-negative patients (8.2891 vs. 11.4262 IU/mL, respectively) [28]. These findings may be explained by the assumption that vitamin D deficiency can cause a persistent HPV infection and thus lead to the development of CIN [29]. A high intake of vitamin D may therefore suppress persistent HPV infection and prevent the development of CIN 1.

### 2.3. Carotenoid

Carotenoids are pigments in plants, algae, and photosynthetic bacteria that produce the bright yellow, red and orange colors seen in plants, vegetables and fruits. There are more than 600 different types of carotenoids. Some can be converted into vitamin A (e.g., alpha carotene, beta carotene, and beta cryptoxanthin) when released into the body. Lutein, zeaxanthin, and lycopene are among the most common carotenoids. 

A total of 433 women participating in the Ludwig-McGill HPV Natural History Study were evaluated using a nested case–control design [30]. In comparison to women who persistently tested positive for HPV, those with transient HPV infection had a higher daily mean intake of lutein/zeaxanthin (296 vs. 249 μg; *p* = 0.03). In a prospective cohort study of 201 women, the intake of lutein was also reported to be associated with a 50–63% reduction in the risk of HPV persistence [31]. In a prospective study designed to investigate the natural history of any type of HPV infection among young women, the adjusted OR for women with a low dietary intake of combined carotenoids (i.e., lutein + zeaxanthin, alpha-carotene, beta-carotene, beta-cryptoxanthin, and lycopene) was 2.4 (95% CI = 1.1–5.2) [32]. 

Regarding the clearance of an oncogenic HPV infection, the adjusted hazard ratios of the highest tertiles of trans- and cis-lycopene in comparison to the lowest tertiles in a prospective study were 2.79 (95% CI = 1.17–6.66) and 2.92(95% CI = 1.28–6.63), respectively [33]. In a Hawaiian cohort study of 122 women, higher circulating levels of trans-zeaxanthin, total trans-lutein/zeaxanthin, cryptoxanthin, trans-lycopene, and cis-lycopene were associated with a significant decrease in the clearance time of type-specific HPV infection, particularly during the early stage of infection (≤120 days) [34]. 

However, in a case–control study in Sao Paulo, low serum levels of lycopene tended to increase the risk of developing CIN 3, whereas medium-to-high levels reduced the risk of developing CIN3 [35,36,37]. Carotenoids may mainly inhibit HPV infection, and lycopene may inhibit the development of CIN 3. 

### 2.4. Folate

Folate (vitamin-9) plays an important role in red blood cells, DNA synthesis, DNA repair, DNA methylation, and cell proliferation [38]. In a cohort of 345 women, Piyathilake et al. showed that a higher folate level showed a significant inverse association with positivity for high-risk HPV infection [38]. They suggested that, in subjects with high folate levels, persistent HPV infection may be decreased and clearance of HPV may be increased as a result of folate preventing the integration of HPV.

In a case–control study of 485 women, Hernandez et al. reported that total folate showed an inverse, dose-responsive association with both low-grade squamous intraepithelial lesions (LSILs) and high-grade squamous intraepithelial lesions (HSILs) [39]. In their case–control study, Kwanbunjan et al. reported that serum folate levels in low-grade (*p* < 0.01) and high-grade cervical dysplasia (*p* < 0.01) cases were markedly lower in comparison to control women [40]. In addition, Tomita et al. suggested that polymorphisms in genes related to folate metabolism modify the association of dietary and circulating folate and vitamin B-6 with cervical dysplasia [41]. Thus, folate may inhibit HPV infection and various grades of CIN.

### 2.5. Vitamin C

Vitamin C, also known as ascorbic acid, has several important functions that help protect and maintain the health of cells, maintaining healthy skin, blood vessels, bones, and cartilage while also helping with wound healing [42]. Vitamin C deficiency can lead to scurvy.

In a nested case–control study of 433 women, the risk of persistent HPV infection was associated with a low intake of vitamin C (adjusted OR, 0.50: 95% CI, 0.27–0.92 (highest vs. lowest quartile)) [30]. In a meta-analysis (one prospective cohort study and 11 case–control studies), the vitamin C intake was also significantly associated with a reduced risk of cervical neoplasia (from CIN to cervical cancer) (OR = 0.58; 95% CI: 0.44–0.75; *p* < 0.001) [43]. Furthermore, an increase in the vitamin C intake by 50 mg/day was related to a reduced risk of CIN (OR = 0.92; 95% CI: 0.89–0.94; *p* < 0.05), indicating dose dependence.

In addition, Manju et al. demonstrated that vitamin C levels in cervical cancer patients were significantly lower in comparison to controls [44]. However, treatment with cisplatin (CDDP) and vitamin C ameliorated the induction of cell death by the overexpression of p53 and generation of hydrogen peroxide in culture cells (SiHa cells), thereby reducing the dose of CDDP required to induce cell death in cancer cells [45]. Vitamin C may therefore reduce HPV infection and inhibit the development of CIN and cervical cancer.

### 2.6. Vitamin E (Tocopherol)

Vitamin E (tocopherol) is a fat-soluble antioxidant that stops the production of ROS that occurs when fat undergoes oxidation. Scientists are investigating whether or not vitamin E may help prevent or delay the onset of chronic diseases associated with free radicals [46]. Vitamin E has also been reported to protect cells from oxidative DNA damage and mutagenesis, thereby preventing the development of some tumors [47].

In a cohort study, tocopherol was reported to be protective against non-oncogenic HPV persistence, but not type-specific oncogenic HPV persistence [48]. The mechanisms involved are unclear, but tocopherols have been suggested to protect against HPV persistence by enhancing the immunological functions and modulating the inflammatory response to infection [49]. 

In a cross-sectional study of 230 women, however, the mean plasma levels of alpha and gamma tocopherol in patients with various grades of CIN and cervical cancer were significantly lower in comparison to controls (*p* < 0.001 and <0.001, respectively, by the Kruskal–Wallis test) [50]. Manju et al. also found that the level of vitamin E in cervical cancer patients were significantly lower in comparison to controls [40]. As above, vitamin E may widely inhibit HPV infection as well as the development of CIN and cervical cancer.

### 2.7. Polyphenols

Polyphenols are the most abundant antioxidants consumed by humans, with a total intake as high as 1 g/day [51]. Plant polyphenols can be divided into several classes, generally depending on the number of phenol rings contained in the structure [52]. These compounds can be divided into two main groups: flavonoids and non-flavonoids. Flavonoids are the largest polyphenolic compounds. This group of compounds is divided into six major subclasses: flavonols, flavones, isoflavones, flavanones, anthocyanidins, and flavanols. 

In the most recent studies, the anti-cancerogenic activities of the most commonly studied natural polyphenols used for the prevention and treatment of cervical cancer have been described [53]. Polyphenols inhibit the proliferation of HPV cells through the induction of apoptosis, growth arrest, inhibition of DNA synthesis, and modulation of signal transduction pathways. 

It is important for polyphenols to be used in combination therapy with chemotherapy or radiotherapy against cervical cancers. Several experimental studies have explored this approach. One flavonoid compound that was reported to sensitize cisplatin is wogonin (*O*-methylated flavone), with its main effect shown to be synergistic cytotoxicity with the enhancement of apoptotic death in human cervical cancer cells [53,54]. Another adjuvant flavonoid of cisplatin studied in the literature was quercetin (bioflavonoid), a compound that helps sensitize HeLa cells to apoptosis caused by cisplatin [55]. Xu et al. reported on apigenin (4′,5,7-trihydroxy-flavone) and its effects on human cervical cancer cells and concluded that this compound can sensitize HeLa cells to paclitaxel-induced apoptosis through the enhancement of intracellular ROS accumulation [56]. 

Dietary flavonoids and quercetin are independently capable of disrupting increased cervical malignancy by altering the expression of ubiquitin E2S ligase [57]. Alshtwi et al. showed that tea polyphenol enhanced the therapeutic properties of bleomycin [58]. They further reported that tea polyphenol-bleomycin synergistically inhibited uterine cervical cancer cell viability by decreasing proliferation through apoptosis. 

Quercetin is also used to radiosensitize human cervical cancer cells. Lin et al. demonstrated that quercetin can significantly increase tumor radiosensitivity both in vitro and in vivo [59]. Genistein (a soy isoflavone) was reported to be a potent radiosensitizing agent. Shin et al. showed that genistein behaves as a radiosensitizer in the CaSki human cervical cancer cell line, leading to the induction of apoptosis via the modulation of ROS and a decrease in cellular vitality due to the downregulation of the expression of E6 and E7 [60]. As above, polyphenols can be used in combination therapy with chemotherapy or radiotherapy against cervical cancers.

## 3. Foods

### 3.1. Papaya

Papaya (*Carica papaya L.*) is a tropical fruit that is native to the tropics of South America [61]. Papaya is a deliciously sweet fruit with musky undertones and a distinctive pleasant aroma and is rich in vitamins C and A. One serving of papaya can provide approximately 100% of the daily requirement of vitamin C and 30% of the daily requirement of vitamin A. 

A nested case–control study in Sao Paulo reported that consumption of papaya >1 time/week was inversely associated with persistence of HPV infection (adjusted odds ratio (AOR), 0.30; 95% CI, 0.14–0.64) [30]. In addition, a higher reported consumption of papaya was inversely associated with the risk of SIL (*p* = 0.01), with the association strongest when papaya was consumed ≥1 time/week (AOR, 0.19; 95% CI, 0.08–0.49) [62]. Thus, the consumption of papaya may prevent persistent HPV infection from progressing to early dysplastic change. 

### 3.2. Vegetables

The protective properties of vegetables and fruits foods are attributed to the presence of low-molecular antioxidants that protect human cells and their structures against oxidative damage [63]. The effect of reducing the risk of many diseases is not only due individual antioxidants, such as vitamins, but may also potentially be influenced by as yet unidentified antioxidant compounds or the synergistic effects of several different antioxidants present in vegetables and fruits. 

Vegetables include dark green-, dark yellow-, and dark orange-colored vegetables as well as cruciferous vegetables. In a prospective cohort study, Sedjo et al. showed that greater vegetable consumption was associated with a 54% reduction in the risk of HPV persistence (AOR, 0.46; 95% CI, 0.21–0.97) [31]. 

In cross-sectional population-based cervical cancer screening studies in China, the comparison of the lowest and highest tertiles regarding the consumption of onion-related vegetables and legumes yielded AORs of 0.589 (95% CI, 0.387–0.897, *p* = 0.011) and 0.591 (95% CI, 0.392–0.892, *p* = 0.012), respectively, for the risk of CIN 2 among 748 HPV-positive women [64]. Hwang et al. evaluated the relationship between the dietary intake of vegetables and the risk of CIN and investigated whether or not these associations were modified by the HPV viral load [65]. Subjects with a decreased intake of vegetables and an increased viral load (≥15.5) had a higher risk of developing CIN 2/3 (OR = 2.84; 95% CI, 1.26–6.42, *p* value for interaction = 0.06 for vegetables) in comparison to those with a decreased intake of vegetables and a decreased viral load (<15.5). Furthermore, a low intake of whole vegetables was not protective against persistent HPV infection or CIN 1, 2, or 3 [37]. Jia et al. also reported that more fresh vegetables in the diet decreased the risk of CIN 2/3 and cervical cancer (OR = 0.896; 95% CI, 0.809–0.993, *p* = 0.035) in a case–control study [66]. In brief, the frequent consumption of vegetables may prevent the development of cervical cancer, starting with HPV infection. 

### 3.3. Fruits

Fruits include many low-molecular-weight antioxidants. The effects of fruit consumption on cervical cancer development have been reported to be similar to those of vegetables.

Barchitta et al. reported that moderate adherence to a Mediterranean diet, which includes fruits and legume vegetables in addition to fats and oils, such as olive oil and fish oil, decreased the odds of HR-HPV infection in comparison to low adherence (AOR = 0.40; 95% CI, 0.22–0.73) in a cross-sectional study [67]. In addition, a low intake of fruits and juices showed a marginal inverse association with HPV persistence (AOR = 0.46; 95% CI, 0.19–1.06) [31].

In a nested case–control study of 265 HPV-positive women, the risk of SIL was reduced among those who reported consuming oranges ≥1 time/week (AOR = 0.32; 95% CI, 0.12–0.87, *p* = 0.02) [62]. As with vegetables, patients with a decreased intake of fruits and an increased viral load (≥15.5) had a higher risk of developing CIN 2/3 (OR = 2.93; 95% CI, 1.25–6.87, *p* value for interaction = 0.01 for fruits) in comparison to patients with a decreased intake of fruits and a decreased viral load (<15.5) [65]. González et al. also observed a statistically significant inverse association between invasive cervical cancer and a daily intake of 100 g of total fruits (hazard ratio = 0.83; 95% CI, 0.72–0.98) [68]. The frequent consumption of fruits may prevent the process of cervical cancer development, starting with HPV infection. 

## 4. Discussion

The effects of individual nutrients and foodstuff on cervical cancer development (HPV infection; CIN 1, 2, and 3; and cervical cancer) are summarized in Figure 1.

Regarding nutrients, different antioxidants may have differing abilities to intervene in the natural history of cervical diseases associated with HPV infection. The intake of vitamins A and D may inhibit early events of cervical cancer development (from HPV infection to CIN 1 development) [14,16,17,18,26,27,28,29]. In addition, the intake of carotenoids may also inhibit early events (HPV infection) [30,31,32,33,34]. However, the carotenoid lycopene was also reported to inhibit CIN 3 development [35,36,37]. The reason for lycopene’s multiple mechanisms of action is unclear. The intake of folate was reported to potentially inhibit the events from HPV infection to the development of various grades of CIN [38,39,40]. Furthermore, the intake of vitamins C and E may widely inhibit cervical cancer development (from HPV infection to the development of CIN 1, 2, and 3 as well as cervical cancer) [30,43,44,45,48,49,50].

Nutrient intake likely cannot inhibit cervical cancer growth. The suppressive effects of vitamins C and E against cervical cancer are suspected to result from the inhibition of HPV infection and cervical dysplasia or to occur with chemotherapy in patients who received combination therapy. Polyphenols are often used in the treatment of cervical cancer in combination with chemotherapy and radiation therapy [53,54,55,56,57,58,59,60]. The high-dose intake of polyphenols, such as isoflavone, has been reported to improve the menopause-specific quality of life (MENQOL), too [69,70].

Regarding foodstuffs, both vegetables and fruits contain multiple vitamins. 

There have been reports with clinical data, indicating that consuming these foods may widely suppress the development of cervical cancer (from HPV infection to the development of CIN and cervical cancer) [31,37,62,64,65,66,67,68]. There are many kinds of vegetables and fruits and the ratios of antioxidants contained in each vegetable and fruit are different. The effects of each vegetable and fruit on cervical diseases are thus expected to show slight differences.

The present study reviewed the relationship between the dietary and nutrient intake and the cervical cancer risk. Most previous papers on the subject have been epidemiological studies with clinical data; few experimental studies have been reported. Thus, the mechanisms through which diet and nutrition influence cervical cancer development are unclear. The mechanisms through which these nutrients prevent the development of cervical cancer should be examined in greater detail using an experimental design. 

Accelerator and brake roles exist in cervical cancer development. The above-mentioned dietary antioxidants function as brakes on cervical cancer development. Accelerators, in contrast, include environmental, immunological, and lifestyle cofactors such as cigarette smoking, diet, oral contraceptive (OC) use, parity, and coinfection with other sexually transmitted infections [10]. Indeed, we previously reviewed the effect of cigarette smoking on the risk of cervical cancer development [71,72], hypothesizing that the promotive effect of smoking may be stronger than the preventive effect of antioxidant intake against cervical cancer. Wei et al. suggested that in HR-HPV-infected cells, tobacco smoking not only causes DNA adducts and strand breaks [73], but also causes an increase in the viral load [74]. Tobacco smoking induces the heightened expression of E6 and E7, which has further adverse effects on the control of the cell cycle, DNA damage repair, and protective apoptosis; thus, cervical cells continue to accumulate mutations that permit malignant transformation [74]. Based on the findings of the present study, healthcare workers should advise women with HPV pre-infection or infection to increase their intake of dietary antioxidants and to avoid smoking with the aim of preventing the development of cervical cancer. It is important for women to have a wealth of knowledge on the factors that have accelerator and brake roles on cervical cancer development. Healthcare workers should follow these patients using specific biomarkers [75,76].

## 5. Conclusions

Regarding nutrients, different antioxidants may have differing abilities to intervene in the natural history of cervical diseases with HPV infection. In particular, each vitamin may have different suppressive effects in different stages of cervical cancer development (from HPV infection to the development of CIN and cervical cancer). In contrast, the intake of vegetables and fruits containing multivitamins may widely suppress cervical cancer development. Healthcare workers should therefore advise women with HPV pre-infection or infection to increase their intake of dietary antioxidants and to avoid smoking with the aim of preventing the development of cervical cancer. 

## Figures and Tables

**Figure 1 medicina-56-00604-f001:**
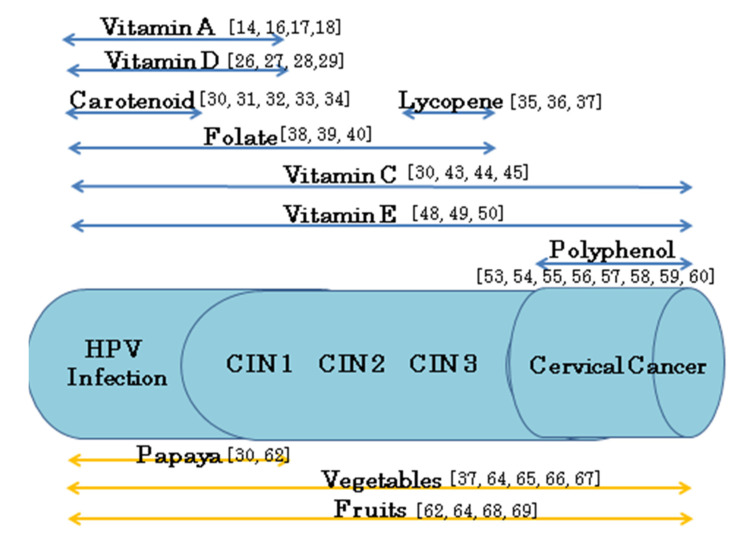
The effects of individual nutrients and foodstuffs on the development of cervical cancer.

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
