# Peer review of "The Preventive Effect of Dietary Antioxidants on Cervical Cancer Development"

_medicina, 2020, doi:10.3390/medicina56110604_

Round 1
Reviewer 1 Report
I read with great interest the Manuscript titled “The preventive effect of dietary antioxidants on cervical cancer development” (medicina-970252), which falls within the aim of Medicina. In my honest opinion, the topic is interesting enough to attract the readers’ attention. Nevertheless, authors should further improve the Manuscript.
Authors should consider the following recommendations:
· Manuscript should be further revised by a native English speaker in order to improve readability and correct several typos.
· It would be interesting to discuss the role of nutrients in other gynecological conditions such as menopause. Some interesting articles on this topic are: PMID: 29725283; PMID: 19411814
· To date, several lines of evidence support the possibility to use specific biomarkers to identify early stage cervical cancer and, in this way, offer a better prognosis to the patients. This point deserves to be discussed, referring to: PMID: 29787011; PMID: 30579259.
Author Response
Answer to the referees’ comments.
Corrected parts were shown with red color in Text.
Authors should consider the following recommendations:
Manuscript should be further revised by a native English speaker in order to improve readability and correct several typos.
*Thank you very much for your advice. Yes, a native English speaker carefully checked our article again. The corrected parts are shown with red color.
It would be interesting to discuss the role of nutrients in other gynecological conditions such as menopause. Some interesting articles on this topic are: PMID: 29725283; PMID: 19411814
*Thank you very much for your good idea. Yes, we inserted the following sentence in Discussion; The high-dose intake of polyphenols, such as isoflavone, has been reported to improve the menopause-specific quality of life (MENQOL), too [70, 71].
To date, several lines of evidence support the possibility to use specific biomarkers to identify early stage cervical cancer and, in this way, offer a better prognosis to the patients. This point deserves to be discussed, referring to: PMID: 29787011; PMID: 30579259.
*Thank you very much for your good idea. Yes, we inserted the following sentence in Discussion; It is important for women to have a wealth of knowledge on the factors that have accelerator and brake roles in cervical cancer development. Healthcare workers should follow these patients using specific biomarkers [76, 77].

Reviewer 2 Report
This review by Ono et al addresses the role of dietary antioxidants on cervical cancer development. While this review is important, there are several aspects of nutrient antioxidants that should also be also be considered. Here are my comments:
[1] Cervical cancer cases dominate in poorer countries where availability of even three proper meals a day is a challenge. Availability of fresh fruits like oranges may be out of reach for such populations. Can the authors address this global problem ? Are there any current considerations in different countries that address this problem ?
[2] Pills in form of vitamins and minerals could be a cheap source to provide nutrients in affected populations where fresh fruits and vegetables are not easily available. Are there any efforts by countries to provide such means to their populations as a healthcare measure ?
[3] The review does not address the role of cigarette smoking and use of tobacco related products that could counteract the benefit of a healthy diet if food sources are available. Can the authors address this issue.
[4] The authors use the word "might" throughout the manuscript. This is not incorrect, but does give the impression that there is no substance to the literature cited and the quality of the review manuscript. It is ok to use the "might" in some places and also to switch between words like "could", "may", "has the potential" or "could potentially".
[5] In some places the < or > signs have a - under them or =. Please check these.
[6] The authors should separate the discussion of fruits and vegetables into two sections and expand on each.
[7] The authors mention the Mediterranean diet and that could be expanded to include fats and oils, like olive oil and fish oil and the nutrients that are important in these diets.
Author Response
Answer to the referees’ comments. Corrected parts were shown with red color in Text.
Referee 2
[1] Cervical cancer cases dominate in poorer countries where availability of even three proper meals a day is a challenge. Availability of fresh fruits like oranges may be out of reach for such populations. Can the authors address this global problem ? Are there any current considerations in different countries that address this problem ?
*Thank you very much for your good question. We speculate that there are more people with smoking rather than without intaking of rich fruits in developing country. People in developing countries are not necessarily poor, but lack of knowledge.
[2] Pills in form of vitamins and minerals could be a cheap source to provide nutrients in affected populations where fresh fruits and vegetables are not easily available. Are there any efforts by countries to provide such means to their populations as a healthcare measure ?
*We are sorry that we do not know these systems to distribute pills or supplements in developing countries. We speculate that these costs are relatively high in such countries. Preferably, they should grow fruits and vegetables by themselves. It is economical. What is important is for them to have a wealth of knowledge of cervical cancer development.
[3] The review does not address the role of cigarette smoking and use of tobacco related products that could counteract the benefit of a healthy diet if food sources are available. Can the authors address this issue.
*Thank you very much for your good advice. Yes, we inserted the following sentences in Discussion;
Wei et al. suggested that in HR-HPV-infected cells, tobacco smoking not only causes DNA adducts and strand breaks [74], but also causes an increase in the viral load [75]. Tobacco smoking induces the heightened expression of E6 and E7, which has further adverse effects on the control of the cell cycle, DNA damage repair, and protective apoptosis; thus, cervical cells continue to accumulate mutations that permit malignant transformation [75].
[4] The authors use the word "might" throughout the manuscript. This is not incorrect, but does give the impression that there is no substance to the literature cited and the quality of the review manuscript. It is ok to use the "might" in some places and also to switch between words like "could", "may", "has the potential" or "could potentially".
*Thank you very much. We changed “might” into “may”.
[5] In some places the < or > signs have a - under them or =. Please check these.
*Yes, we checked it.
[6] The authors should separate the discussion of fruits and vegetables into two sections and expand on each.
*Thank you very much for your advice. As mentioned in Discussion, both vegetables and fruits contain multiple vitamins. When we divide the contents into paragraphs of fruits and vegetables, it is not gathered what we want to say. We think that only the nutrients are enough for distributing the work. We are sorry that we cannot change the contents of fruits and vegetables in Discussion.
[7] The authors mention the Mediterranean diet and that could be expanded to include fats and oils, like olive oil and fish oil and the nutrients that are important in these diets.
*Thank you very much for your good advice. Yes, we inserted the following sentence in 3.3 Fruits.
Barchitta et al. reported that moderate adherence to a Mediterranean diet, which includes fruits and legume vegetables in addition to fats and oils, such as olive oil and fish oil, decreased the odds of HR- HPV infection in comparison to low adherence (AOR=0.40; 95% CI, 0.22-0.73) in a cross-sectional study [68].